# The Multifaceted COVID-19: CT Aspects of Its Atypical Pulmonary and Abdominal Manifestations and Complications in Adults and Children. A Pictorial Review

**DOI:** 10.3390/microorganisms9102037

**Published:** 2021-09-26

**Authors:** Chiara Morelli, Mariantonietta Francavilla, Amato Antonio Stabile Ianora, Monica Cozzolino, Alessandra Gualano, Giandomenico Stellacci, Antonello Sacco, Filomenamila Lorusso, Pasquale Pedote, Michele De Ceglie, Arnaldo Scardapane

**Affiliations:** 1Interdisciplinary Department of Medicine, Section of Diagnostic Imaging, University of Bari Medical School, 70124 Bari, Italy; amatoantonio.stabileianora@uniba.it (A.A.S.I.); monicacoz@hotmail.it (M.C.); alegualano@gmail.com (A.G.); sacccoanto@gmail.com (A.S.); milalorusso@yahoo.it (F.L.); pedoterx@gmail.com (P.P.); micheledeceglie@libero.it (M.D.C.); arnaldo.scardapane@gmail.com (A.S.); 2Unit of Pediatric Imaging, Giovanni XXIII Hospital, 70126 Bari, Italy; marianto_fra@hotmail.it (M.F.); gstellacci@icloud.com (G.S.)

**Keywords:** COVID-19, atypical manifestation, CT, adult, children, pulmonary, abdominal

## Abstract

Our daily experience in a COVID hospital has allowed us to learn about this disease in many of its changing and unusual aspects. Some of these uncommon manifestations, however, appeared more frequently than others, giving shape to a multifaceted COVID-19 disease. This pictorial review has the aim to describe the radiological aspects of atypical presentations and of some complications of COVID-19 disease in adults and children and provide a simple guide for radiologists to become familiar with the multiform aspects of this disease.

## 1. Introduction

On 31 December 2019, the Chinese health authority announced an outbreak of pneumonia cases of unknown etiology in Wuhan (Hubei, China). On 9 January 2020, the China Center for Disease Control and Prevention (CDC, Beijiing CHN) identified a new coronavirus (temporarily named 2019-nCoV) as the etiological cause of this disease. They later confirmed the inter-human transmission of the virus. On 11 February, the World Health Organization (WHO, Geneva CH) announced that pneumonia caused by 2019-nCoV was called COVID-19 (Corona Virus Disease 19). The Coronavirus Study Group of the International Committee on Taxonomy of Viruses has classified the virus under the name of SARS-CoV-2, associating it with the coronaviruses that cause severe acute respiratory syndrome (SARS-CoVs, severe acute respiratory syndrome coronaviruses). On 11 March 2020, the WHO, after assessing the severity levels and global spread of the SARS-CoV-2 infection, declared the COVID-19 pandemic. There are 190,770,507 confirmed cases in the world since the beginning of the pandemic and 4,095,924 deaths [1].

While Reverse Transcriptase Polymerase Chain Reaction (RT PCR) remains the diagnostic gold standard of infection, computed tomography (CT) has been shown to have a fundamental role in diagnosis, especially in those patients with false negative RT PCR results, with a sensitivity of approximately 98%. The CT examination allows us to monitor the progression of the disease and evaluate the multi-organ involvement. Thanks to CT, we have learned to recognize the typical pulmonary presentation (which we will describe below). Nevertheless, both the literature data and our direct experience require us to keep in mind that this pathology can often show itself “in disguise”, giving atypical, non-classical, unexpected and complicated manifestations: it is necessary to know them in order to avoid misunderstandings.

Typical SARS-CoV-2 pneumonia is well known and it is characterized by bilateral, multilobar, peripheral, subpleural ground glass opacity [2] (Figure 1). In the most advanced stages of the disease, however, parenchymal consolidations, and thickening of the interlobular septa with crazy paving patterns will be frequently visible [3].

Atypical manifestations may affect both adult and pediatric patients with pulmonary and extra-pulmonary involvement.

## 2. Adult “Atypical” and “Complicated” Pulmonary COVID-19

Lymphadenopathy, pleural effusion, pericardial effusion, bronchiectasis, halo sign and reverse halo sign, cavitation and some complications, such as spontaneous pneumothorax and pneumomediastinum, are reported as either pulmonary unusual manifestations or atypical complications of COVID-19 [4,5,6,7]. As we will clarify below, if lymphadenopathies are more often found in acute phase of the disease, the pleural effusion is frequently identified in the advanced stages. Furthermore, while lymphadenopathies are linked to patients with stronger immune response, pericardial effusion is associated with the worst COVID-19 cases, in which heart damage coexists.

### 2.1. Lymphadenopathy

With the term “lymphadenopathy” we refer to mediastinal lymph nodes >10 mm and hilar lymph nodes >3 mm in short axis diameter. As already reported by Li et al. [8], and by Grassi et al. [9], lymphadenopathies are part of the acute COVID-19 manifestations even if they do not represent a specific and characteristic sign (Figure 2). Xiao Li et al. [10] reported that hilar and mediastinal lymph node enlargement was observed in 43.51% of patients with COVID-19 pneumonia. Their study suggested that enlarged hilar and mediastinal lymph nodes are associated with immune response and that patients with COVID-19 pneumonia have stronger immune response especially in moderate and severe conditions.

### 2.2. Pleural Effusion

Pleural effusion is rarely associated with COVID-19. We often find it in the most critically ill patients and in the most advanced stages of the disease (Figure 3). Zhou [11] et al. described pleural effusion in 3.2% of COVID-19 patients while Grassi et al. [9] observed it in 14.3% of cases. Darwish et al. [12] found that during the first week of infection pleural effusion occurs only in 13.6% of patients. Xiao li et al. [10], in regard to pleural effusion, reported a frequency of 14.3% and concluded that it was not significantly associated with COVID-19 pneumonia.

### 2.3. Pericardial Effusion

Pericardial effusion can be defined as the presence of more than 50 mL of liquid between the pericardial sheets. As already mentioned for the pleural effusion it is not characteristic of the COVID-19 thoracic involvement. In fact, it was observed by Li et al. [8] in 4.8% of cases. However, in this study, it is also described as a manifestation associated with the most compromised patients (Figure 4).

Ali Sabri et al. [13] found a 7.9% frequency of pericardial effusion. Furthermore, from their analysis, it emerged that pericardial effusion could be considered an important factor for admission to intensive care unit (ICU), as it could be an indicator of myocarditis or cardiomyopathy caused by COVID-19, but they did not confirm this hypothesis with echocardiography.

Grassi et al. [9] describe pericardial effusion in 16.7% of patients, proposing a relation with heart damage; Shi et al. [14] associated it with a higher risk of in-hospital mortality.

### 2.4. Bronchiectasis

Although bronchiectasis is not a typical COVID-19 manifestation (as noticed by Salehi et al. [15]), it has been reported with a very high frequency (41.3%) by Auger et al. [16]. They hypothesize that their result is probably due to the particularly compromised clinical status of the patients included in the study. Devie et al. [17] detected bronchiectases and traction bronchiectases in 25.3% of cases, and concluded that these cases were significantly more frequent in the most severe group of patients.

### 2.5. Halo Sign and Reverse Halo Sign

Halo sign consists of a shaded area of increased parenchymal density surrounded by peripheral ground glass changes. It was studied by Wu and Chen et al. [18], who discovered it in 18/130 COVID-19 patients (13%), considering it as an unusual feature associated with the initial phase of the disease. The reverse halo sign can be defined as a clearly rounded area of “ground glass-like” increased parenchymal density, circumscribed by a consolidation ring (Figure 5). It likely represents the disease progression towards consolidation as described by Bernheim et al. [19], who detected it in 4% of the patients studied at 6–12 days of infection.

Kuang et al. [20] conducted a study aimed to identify the CT differences between H1N1 influenza pneumonia and COVID-19. Their results showed that the reverse halo sign in COVID-19 pneumonia was present in 18.5% of cases, and that this sign was more common than in H1N1 influenza pneumonia (*p* < 0.05).

### 2.6. Cavitation

Cavitation is quite rare in viral pneumonia and consequently also in SARS-CoV-2 and MERS-CoV pneumonias [21,22,23].

Zoumut et al. [24] investigated the development of lung cavitation in patients with severe COVID-19 disease treated in ICU; a frequency of 11% was detected. They assumed that cavitation was due to several factors such as bacterial and fungal coinfection, immunosuppressive effects of glucocorticoids and tocilizumab, inflammatory state induced by SARS-CoV-2, and thrombotic diathesis (Figure 6). This study also reported that lung cavitation in patients with severe COVID-19 lung disease can occur and is associated with secondary complications, such as hemoptysis and pneumothorax, conferring a poor prognosis.

### 2.7. Pneumothorax and Pneumomediastinum

As delineated by Vega et al. [25], a cough, typically present in COVID-19 disease, is the cause of these two complications (Figure 7 and Figure 8). They illustrate three cases of spontaneous pneumothorax and pneumomediastinum and argue that, although the development of pneumothorax is usually secondary to barotrauma in patients in ICU, their patients did not require, at diagnosis, assisted mechanical ventilation. They concluded that in their cases the severity of the disease was associated with these rare complications.

Loffi et al. [26] identified 6 cases of spontaneous pneumomediastinum in 102 COVID-19 patients investigated with CT examination (6%). They asserted that spontaneous pneumomediastinum is a possible complication of severe COVID-19 pneumonia and can affect patient management and clinical outcomes. In fact, in their experience “the concomitant presentation of spontaneous pneumomediastinum and diffuse COVID-19 pneumonia was associated with a severe clinical course characterized by sudden ARDS that required aggressive management and early intubation in three patients”.

Table 1 summarizes all the studies reported in Section 2: Adult “Atypical” and “Complicated” Pulmonary COVID-19.

## 3. Children’s “Atypical” and “Complicated” Thoracic COVID-19 Manifestations

As in the adult case series, in pediatric patients COVID-19 pneumonia manifested more frequently with some “typical” features, such as ground glass opacity (GGO), peribronchial thickening, vascular engorgement and airspace consolidation [27,28,29,30]; as stated also by the international consensus [31] regarding pediatric chest imaging. Atypical imaging findings are represented by pleural effusion, atelectasis, nodules, linear opacities, lymphadenopathy, pneumothorax and pneumomediastinum.

### 3.1. Pleural Effusion

Caro-Dominguez et al. [27] found pleural effusion in 7% of their cases, Ugas-Charcape et al. [30] described it in 6% and 13% of chest radiographs and CTs, respectively, while Zhang et al. [29] reported no cases of pleural effusion.

Pleural effusion (Figure 9) was found to be more significantly present in the frame of MIS-C, a pathological multifactorial entity described below.

### 3.2. Atelectasis

The case series reports atelectasis in a few cases: 2% according to Caro-Dominguez et al. [27] and Ugas-Charcape et al. [30]. This feature has been related to the presence of associated pneumothorax and pleural effusion (Figure 9), or to an incorrect position of the endotracheal tube [27].

### 3.3. Nodules and Linear Opacities

In a number of studies “nodules” refer to well-defined ground glass opacities.

Caro-Dominguez et al. [27] described pulmonary nodules in 25% of cases and linear opacities in 33% of chest CT examinations. In a case series from Latin America [30] these findings were reported in 9.7% and 15.6% of pediatric patients, respectively.

### 3.4. Lymphadenopathy

Lymphadenopathy, both hilar and mediastinal, as in the adult series, is an infrequent feature in chest CT examinations. It was categorized as an atypical finding in the international consensus statement of chest imaging in pediatric COVID-19 patients [31]. In a subgroup of studies, it has low prevalence: 17% according to Caro-Dominguez et al. [27] and 18% according to Zhang et al. [29]; other studies did not find any cases of thoracic lymphadenopathy [22,30,32] in children affected by COVID-19.

### 3.5. Pneumothorax and Pneumomediastinum

Spontaneous pneumothorax and pneumomediastinum (Figure 10) are also very infrequent findings in pediatric COVID-19-related pneumonia: a study [27] reported pneumothorax in 2% of cases, and a few case reports in the literature have described these findings [33,34], usually in adolescents with severe disease.

Table 2 summarizes all the studies reported in Section 3: Children’s “Atypical” And “Complicated” Thoracic COVID-19 Manifestations.

### 3.6. Cardiothoracic Mis-C

A hyperinflammatory immune-mediated shock syndrome has been recognized in children aged <19 years were exposed to the severe acute respiratory syndrome coronavirus 2 (SARS-CoV-2). The WHO [35] and United States Center for Disease Control and Prevention (CDC) [36] refer to this entity as Multisystem Inflammatory Syndrome in Children (MIS-C).

The diagnostic criteria of MIS-C have been defined by the WHO [35] and includes biochemical evidence of elevated inflammatory markers, absence of other microbial causes and evidence of organ dysfunction (e.g., hypotension, shock, myocardial dysfunction). At the time of this report more than 4000 cases have been collected worldwide [36].

Affected patients present a wide spectrum of clinical findings consisting of fever, headache, pain at the extremities, abdominal pain, vomiting and diarrhea, skin rash, conjunctivitis and peripheral oedema, with variable severity, with a significant percentage evolving to myocardial damage and cardiogenic, septic or toxic shock [37]. Laboratory data measured in affected children show a marked pro-inflammatory state [38].

The post-viral hyperinflammatory process presumed to cause MIS-C results in unique thoracic imaging abnormalities that differ from the classic manifestations of acute pediatric COVID-19 infection [39].

Although radiological findings are not typical, they may be red flags for the diagnosis of MIS-C when matched with clinical and laboratory data [40].

According to a number of studies the most common X-ray and CT findings are as follows: perihilar opacity and peribronchial thickening; pleural effusion and cardiomegaly [39,40]; atelectasis, airspace consolidation and diffuse ground glass appearance [41,42,43,44].

Cardiac CT and MRI can also show heart failure with left ventricular systolic dysfunction, myocardial oedema, pericardial effusion and coronary artery dilatation [40,41,42,43], the latter in the frame of the so-called Kawasaki-like disease [45], the well-known vasculitis that affects the medium calibre vessels in children.

## 4. Adult “Abdominal” COVID-19

In recent times, the efforts of the international scientific community aimed to define the abdominal manifestations of COVID-19, even if there is a small number of studies published on this topic. Although the pathogenetic mechanisms have yet to be clarified, we know that SARS-CoV-2 enters cells by exploiting the ACE-2 receptor which is widely expressed in the GI tract, pancreas, biliary tract and vascular endothelium [46,47,48]. It is also known that critical patients with COVID-19 have systemic coagulopathy and a thrombotic diathesis supported by the important underlying inflammatory process [48,49,50,51]. Although pulmonary embolism is the most frequent thrombotic complication in these patients, arterial thrombosis could be equally relevant and its presentation as acute aortic occlusion would be evocative [52]. Baeza et al. [53] presented three cases of acute aortic occlusion (AAO) and concluded that despite the pre-existence of risk factors in these patients, there is likely an association between COVID-19 infection and the development of a prothrombotic state leading to significant arterial complications (Figure 11).

This helps to clarify the pathogenetic mechanisms underlying pancreatitis, colitis and abdominal infarction in COVID-19, although further studies are needed.

### 4.1. Kidney, Splenic and Intestinal Infarction

Goldberg-Stein et al. [46] found splenic and renal infarction in 5% of COVID-19 patients investigated with CT examination, related to a state of hypercoagulation.

As described by Bhayana et al. [48] and Goldberg-Stein et al. [46] (reported also by Tirumani et al. [54] and Lui et al. [55]) the pathogenetic mechanism underlying intestinal infarcts in critical COVID-19 patients might be the thrombotic diathesis due to the inflammatory response to the infection. On CT examination we can see mesenteric arterial or venous filling defects, the “paper sheet wall” sign or the “target” sign with thickened wall, constituted by hyperdense mucosa and hypodense submucosa. In the more advanced stages this condition evolves with findings of hydro-aerial levels, and parietal and portomesenteric pneumatosis (Figure 12). Tirumani et al. [54] observed the frequency of intestinal infarction in 1/72 COVID-19 patients investigated with CT examination (1.3%).

### 4.2. Pancreatitis

Because of the expression of ACE-2 receptors also in pancreatic cells, pancreatitis can occur in COVID-19 patients [46,47,48] (Figure 13). Funt et al. [47] investigated the presentation rate of pancreatitis in COVID-19 patients examined with CT, which amounted to 1.5%. They searched for the most common causes of abdominal pain in two groups of patients: COVID-19 positive and COVID-19 negative, respectively. In the subset of acute disease, they documented: inflamed bowel, pancreatitis, pyelonephritis or cystitis more frequently in COVID-19+ patients rather than in COVID-19- patients. As illustrated by Wang et al. [56], pancreatitis presents in COVID-19 with a variable percentage ranging from 1–2% of mild cases to 17% of severe cases. Bozdag A. et al. [57] argue that in the literature the pancreatitis diagnosis was based only on amylase and lipase elevation, while the radiological findings were described only in case reports. In one case report necrotizing pancreatitis was described [58].

### 4.3. Colitis, Enteritis

Goldberg-Stein et al. [46] reported that the abnormalities of the gastrointestinal tract were the most common extra-pulmonary CT manifestations in COVID-19 patients. This result was in agreement with those referred by Bhayana et al. [48], who found that 29% of CT scans showed intestinal wall thickening involving the colon or small intestine. The frequency of wall thickening in the gastrointestinal tract in the study by Goldberg Stein [46] was lower (15%); they justify this difference by the fact that the intestinal wall thickening may be relatively underestimated in the clinical setting compared to the experimental one (Figure 14).

### 4.4. Cystitis and Cholecystitis

Goldberg-Stein et al. [46] and Funt et al. [47] reported cystitis in 5% and 4,1% of COVID-19 cases, respectively, as an edematous thickening of the bladder wall with an hyperdense aspect of the mucosa after contrast medium administration (Figure 15). In the Goldberg-Stein et al. [46] paper, CT abnormal findings related to the gallbladder and biliary system were reported in 25% of cases, including distention of the gallbladder, mural oedema, and findings reported as possible or definite acute cholecystitis (Figure 15); 10% of patients had bile duct dilatation. In two studies it is explained that there may be an unclear SARS-CoV-2-induced cytopathic effect on hepatocytes, or an ACE-2 receptor-mediated direct viral infection [59,60]. Since gallbladder wall oedema is a common finding in acute hepatitis, and is an independent predictor of a more severe clinical course [61], it is possible that gallbladder wall oedema seen in some of the patients is a reflex of hepatocellular damage, either directly induced by SARS-CoV-2, or through an inflammatory response. Further studies are needed to determine the link between COVID-19 positivity and biliary and gallbladder pathologic implication.

Table 3 summarizes all the studies reported in the Section 4: Adult “Abdominal” COVID-19.

## 5. Pediatric “Abdominal” COVID-19 Manifestations

Abdominal manifestations of COVID-19 in children include mainly gastrointestinal (GI), hepatobiliary and pancreatic involvement. Multisystem inflammatory syndrome in children (MIS-C) is a complex entity that usually involves the gastrointestinal tract.

### 5.1. Gastrointestinal Manifestations

As explained above, SARS-CoV-2 enters cells via the angiotensin-converting enzyme-2 (ACE-2) receptor, which is abundantly expressed on lung cells, but also on many extra-pulmonary tissues, including gastrointestinal (GI) tract, heart, liver, and kidney [62].

A number of studies have tried to estimate the prevalence of gastroenteric involvement in COVID-19 infection in children, which is matter of debate: Miller et al. [63] showed that gastrointestinal manifestations were present in 84.1% of children admitted to the hospital and were more often associated with fever and rash; Akobeng et al. [64] in a meta-analysis including 280 children from 9 studies estimated that the pooled prevalence of gastrointestinal manifestations was 22.8%, with diarrhea as the most common presentation (12.4%), followed by vomiting (10.3%) and abdominal pain (5.4%).

GI signs and symptoms appear characteristically as presenting features of SARS-CoV-2-related multisystem inflammatory syndrome in children (MIS-C), a condition described below [63].

Little is known about typical imaging findings in COVID-19 pediatric patients with GI symptoms. Abdominal ultrasonography, computed tomography or magnetic resonance imaging may be taken into consideration in patients with a severe course of disease or relevant blood test alterations. Miller et al. [63] collected images from 15 patients with GI symptoms, finding: mesenteric adenitis, biliary sludge or acalculous cholecystitis, and ascites. In three patients, ultrasonography or magnetic resonance imaging showed bowel wall thickening.

Tullie et al. [65] found on abdominal ultrasound the presence of lymphadenopathy, inflammatory fat throughout the mesentery, and thickening of the terminal ileum, that were confirmed with CT examination when performed.

Less frequent manifestations of GI SARS-CoV-2 involvement are reported in the literature, consisting mainly of acute appendicitis, phlegmonous ileocolitis, intussusception, pneumatosis intestinalis and protein losing enteropathy [66].

### 5.2. Hepatobiliary and Pancreatic Involvement

A study demonstrated that the distribution of ACE-2 is peculiar; it is highly expressed in the endothelial layer of small blood vessels but not in the sinusoidal endothelium [67]. Its concentration on the surface of cholangiocytes is higher than of the hepatocyte surface and is similar to the type II lungs alveolar cells [68]. SARS-CoV-2 may have the ability to infect cholangiocytes via the ACE-2 receptor and directly dysregulate liver function [67]. Moreover, the induced intestinal inflammation impairs the intestinal mucosal barrier, allowing easy access to the circulation and the possibility to reach and affect other organs, including the liver [69].

Hepatitis was defined as an elevation of alanine aminotransferase (ALT) >40 and aspartate aminotransferase (AST) >50, as these values fall above the 97° percentile for all ages and both sexes, as defined by Bussler et al. [70].

A mild increase in liver enzymes is well described in COVID-19 pediatric patients, with various results among studies, ranging from 13% to 50%; however, serious liver dysfunction is uncommon [71]. Elevated aspartate aminotransferase levels (>50 UI/L) are observed more frequently than alanine aminotransferase levels (>45 UI/L) [72].

Cantor et al. [73] described abdominal imaging performed on patients with hepatitis. Abdominal ultrasounds showed abnormal liver-associated manifestation such as: significant ascites, hepatomegaly or a thick-walled gallbladder. One of the two magnetic resonance imaging studies showed ascites.

SARS-CoV-2 can affect both the exocrine and endocrine pancreas; an abnormal elevation of amylase and lipase, together with glucose dysregulation and acute diabetes, are described in pediatric patients with severe COVID-19 pneumonia, with development of acute pancreatitis [74]. 

A few cases of pancreatic involvement have been reported in the literature in pediatric patients. Samies et al. [75] illustrated three cases of pancreatitis in children affected by COVID-19 that were diagnosed by laboratory tests and imaging (ultrasound or CT examinations). 

Both hepatitis and pancreatitis are observed more frequently in association with MIS-C, as displayed below.

Table 4 summarizes all the studies reported in Section 5: Pediatric “Abdominal” COVID-19 Manifestations.

### 5.3. MIS-C

The multisystem inflammatory syndrome in children (MIS-C), already mentioned above [35,36,37,38], reported in patients under 19 years old who have a history of exposure to SARS-CoV-2, has features that extensively involve the abdomen.

In regard to abdominal MIS-C, Caro-Dominguez et al. [38] found that the most common abnormalities in US, CT and MRI were free fluid (71%) and terminal ileum wall thickening (57%) (Figure 16); other less common findings were hepatomegaly, right iliac fossa lymphadenopathy, gallbladder wall oedema, gallbladder sludge, periportal oedema, splenomegaly and haemorrhagic cystitis. According to Palabiyik et al. [40] the most conventional US and CT findings were hepatomegaly and hepatosplenomegaly, followed by periportal and pericholecystic wall oedema, mesenteric lymph nodes in the right lower quadrant, free fluid in the abdomen, temporary invagination, echogenic kidneys and a case of pancreatic alterations. Additionally, in the series of Hameed et al. [43], the most common US and CT abdominal findings were represented by anechoic free fluid (53%), localized inflammatory change within the right iliac fossa (47%), a combination of echogenic expanded mesenteric fat (37%), and multiple mildly enlarged lymph nodes in 47% of cases.

## 6. Conclusions

In conclusion, this pictorial review shows the multiple aspects of COVID-19 infection which make it a multi-organ disease. These aspects must be known and investigated to obtain the best stratification and ensure optimal treatment for the patient. In fact, although lymphadenopathies, pleural and pericardial effusion, bronchiectases, halo sign, reverse halo sign and cavitations are notoriously associated with bacterial and fungal pneumonia, our review aims to underline (through the scientific data reported), that these manifestations are found in a non-negligible percentage of COVID-19 cases. Furthermore, these atypical thoracic and gastrointestinal manifestations are associated with more severe clinical settings, with systemic involvement and poor prognosis. We believe that these findings can give a real clinical contribution by helping the clinicians to recognize COVID-19 “red flags” associated with worse scenarios. CT is the best imaging method in this environment, because it can be used to assess the progression of the disease and the multisystem involvement.

## Figures and Tables

**Figure 1 microorganisms-09-02037-f001:**
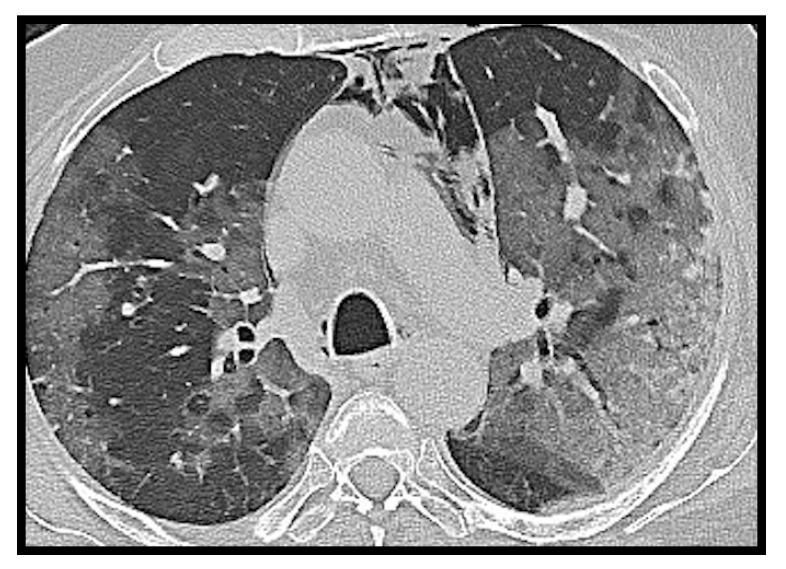
Computed tomography (CT) axial image of ground glass opacity. A hospitalized COVID-19 patient with fever and dyspnea. The axial CT image shows bilateral ground glass opacities with peripheral subpleural disposition in the right lung and tending to confluence in the left lung. Spontaneous pneumomediastinum is associated.

**Figure 2 microorganisms-09-02037-f002:**
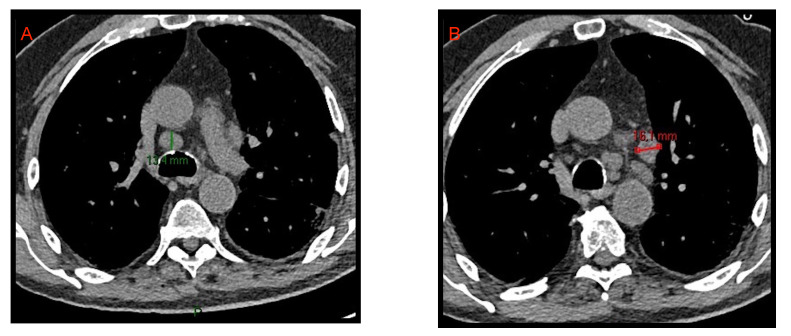
CT axial images of mediastinal lymphadenopathies. Enlarged lymph nodes with short axis of 13.4 mm and 16.1 mm are evident in the lower pre-tracheal space (**A**) and in the aorto-pulmonary window (**B**), respectively.

**Figure 3 microorganisms-09-02037-f003:**
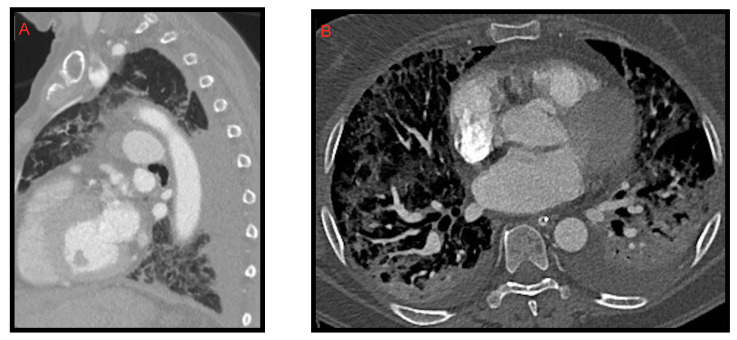
A 58-year-old female patient with worsening respiratory function on the 10th day of hospitalization evaluated with CT. Sagittal (**A**) and axial (**B**) images show posterior pleural effusion greater on the left side.

**Figure 4 microorganisms-09-02037-f004:**
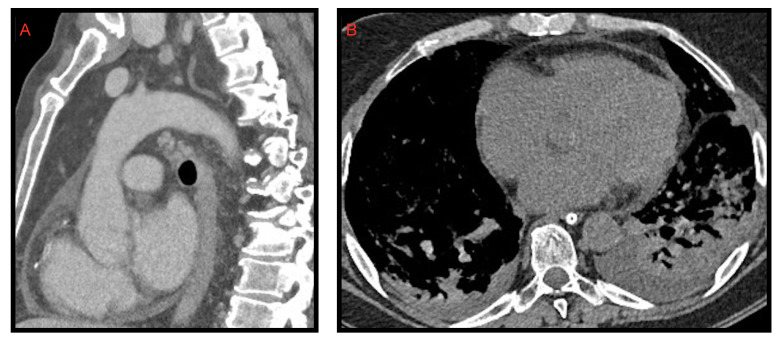
Pericardial effusion on CT, sagittal (**A**) and axial (**B**) images, in two patients admitted to intensive care unit. In image B, bilateral pleural effusion is also noted.

**Figure 5 microorganisms-09-02037-f005:**
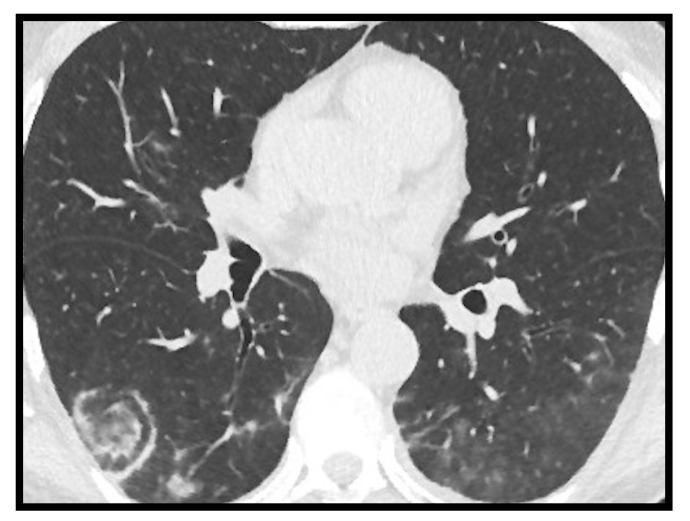
Reverse halo sign in right lower lobe in COVID-19.

**Figure 6 microorganisms-09-02037-f006:**
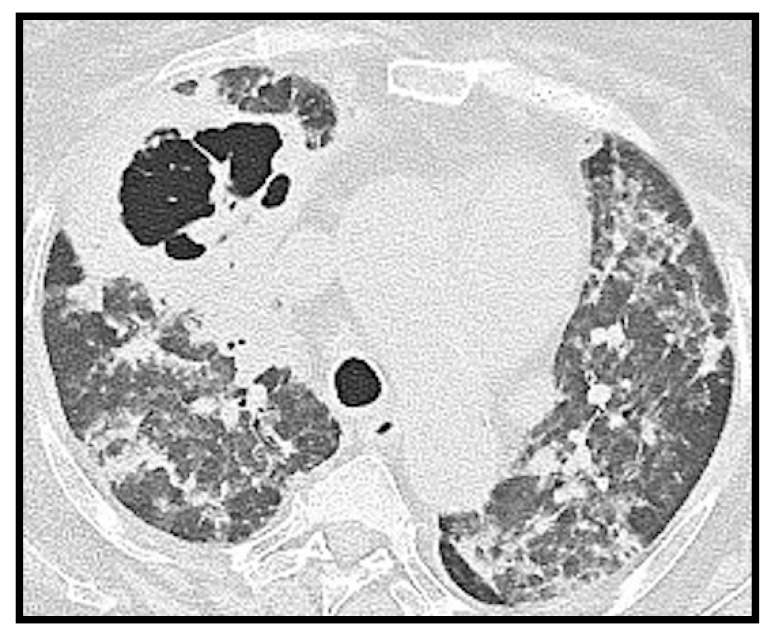
Axial CT image of a 70-year-old female patient. Gross parenchymal consolidation with central excavation suspected for superinfection is evident in the anterior segment of the right upper lobe. In addition, reinforcing suspicion of superinfection, the study was negative for pulmonary embolism.

**Figure 7 microorganisms-09-02037-f007:**
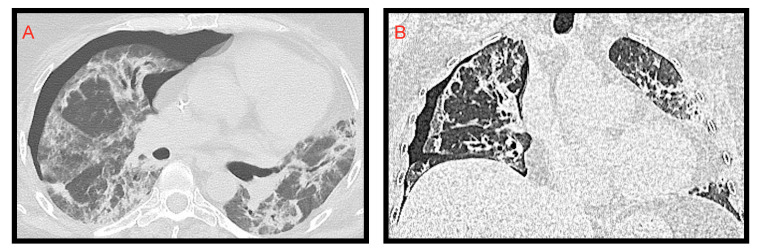
Spontaneous right pneumothorax extended from apex to lung base. Axial (**A**); coronal (**B**); the right lung appears hypo-expanded.

**Figure 8 microorganisms-09-02037-f008:**
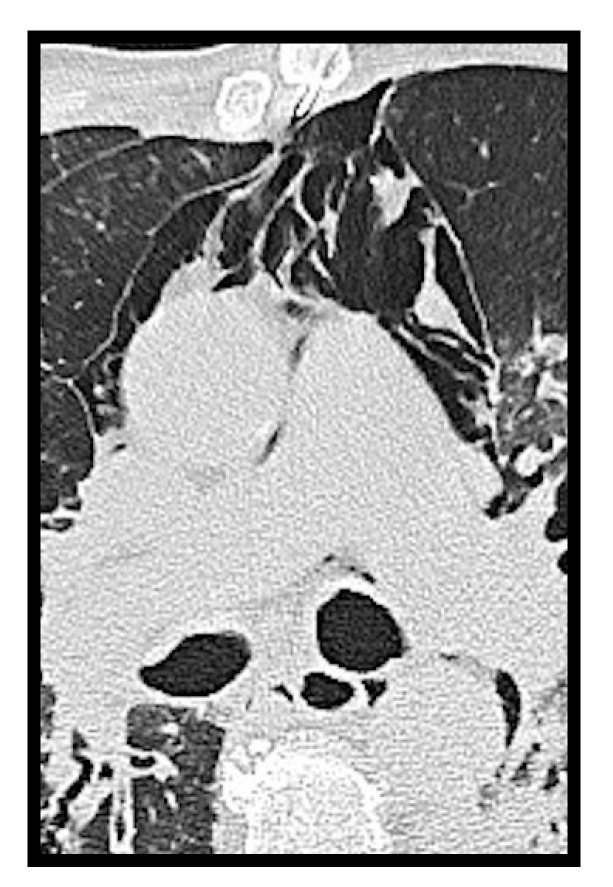
Spontaneous pneumomediastinum in a COVID-19 patient.

**Figure 9 microorganisms-09-02037-f009:**
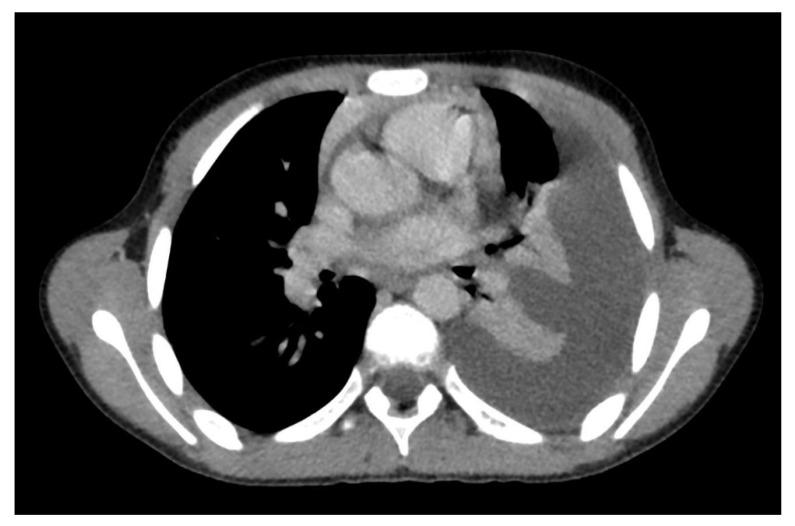
Pleural effusion of the left lung with associated pulmonary atelectasis in a 10-year-old boy.

**Figure 10 microorganisms-09-02037-f010:**
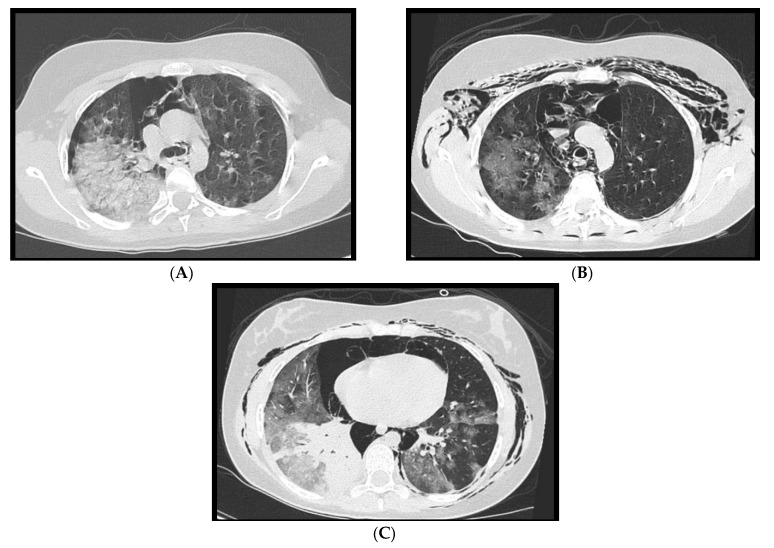
(**A**–**C**): A 14-year-old girl with spontaneous pneumothorax and pneumomediastinum, in association with extensive ground glass changes. In Figure 10C consolidation of the right lower lobe is noted.

**Figure 11 microorganisms-09-02037-f011:**
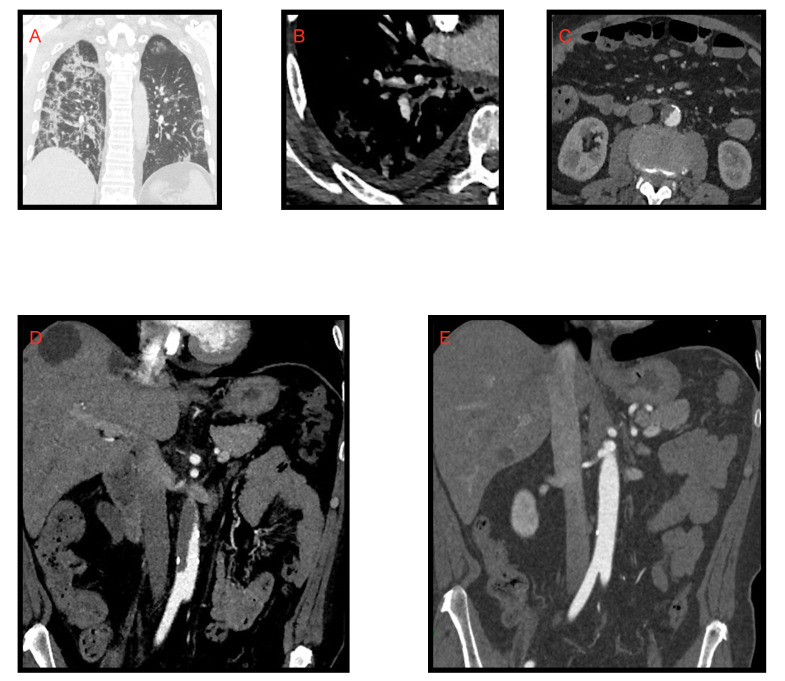
Case of acute aortic parietal thrombosis in patient affected by SARS-CoV-2 pneumonia. Image (**A**) shows the lung’s involvement caused by COVID-19 pneumonia. Thromboembolic opacification defect of a segmental (anterior) arterial branch for the right lower lobe is associated (**B**). (**C**) Axial images of subrenal acute aortic thrombosis. In coronal image (**D**) is it possible to notice the acute aortic thrombosis which extends craniocaudally for about 6 cm. (**E**) Complete resolution of aortic thrombosis after therapy.

**Figure 12 microorganisms-09-02037-f012:**
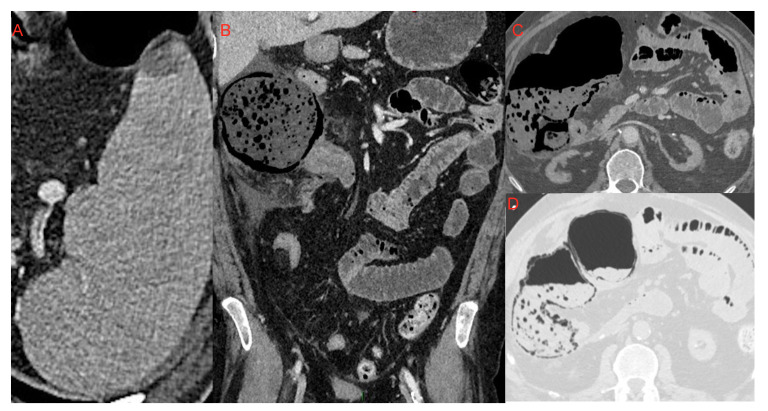
Small area of splenic infarction (**A**); infarction of the cecum and right colon (**B**–**D**): It is possible to notice the marked dilatation of the ascending colon, with parietal pneumatosis and hydro-aerial levels. It was associated with abdominal effusion in right subhepatic space and paracolic gutter which extended to the pelvic cavity.

**Figure 13 microorganisms-09-02037-f013:**
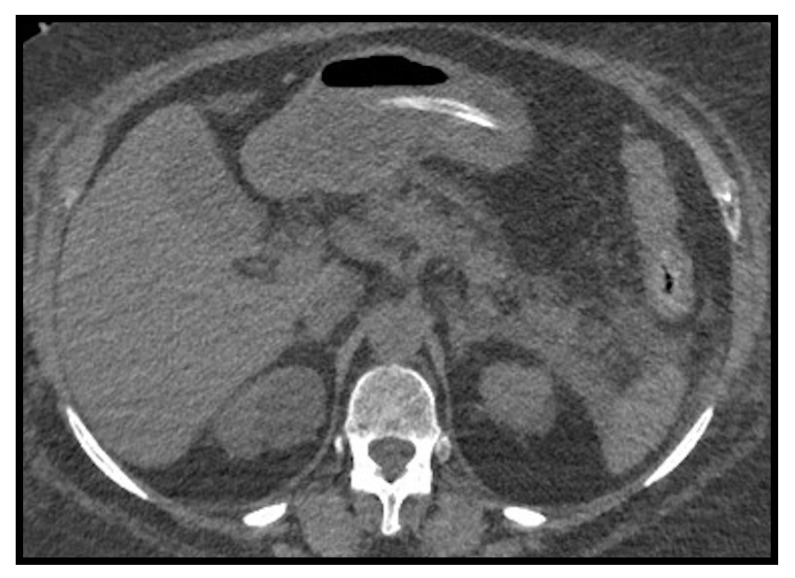
Pancreatitis. Enlarged and edematous pancreas especially at the head, with fat stranding and peripancreatic fluid collection extending along the left anterior pararenal space.

**Figure 14 microorganisms-09-02037-f014:**
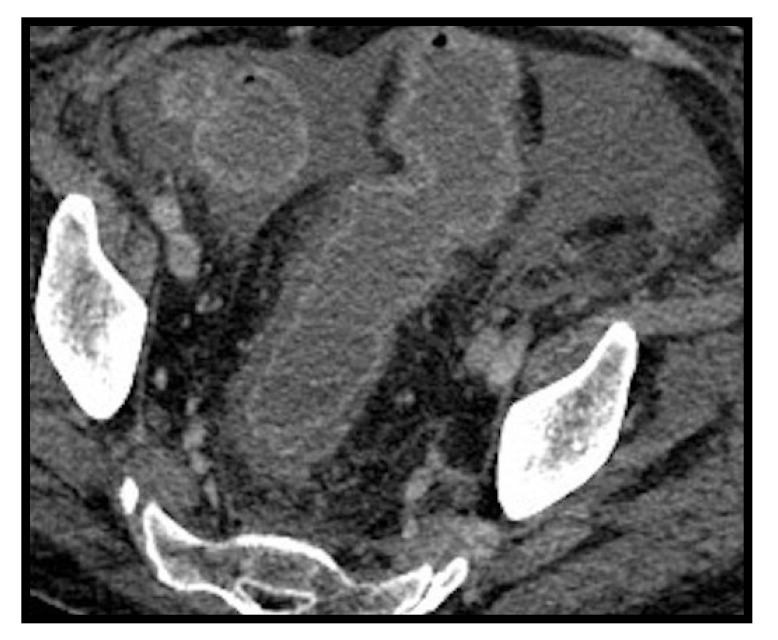
Colitis. Edematous thickening of the walls of the rectum-sigma with hyperdense aspects of the mucosa showing contrast enhancement after administration of contrast medium. Intra-abdominal effusion is associated.

**Figure 15 microorganisms-09-02037-f015:**
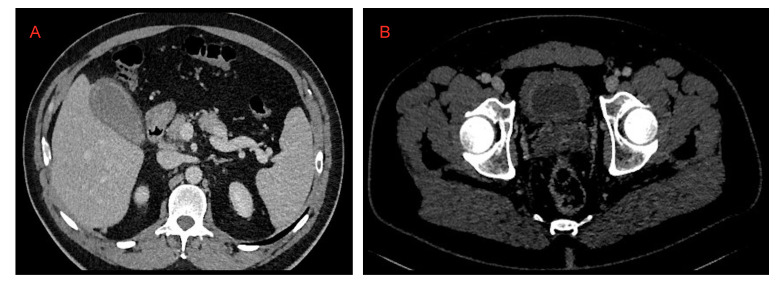
(**A**) Cholecystitis and (**B**) cystitis.

**Figure 16 microorganisms-09-02037-f016:**
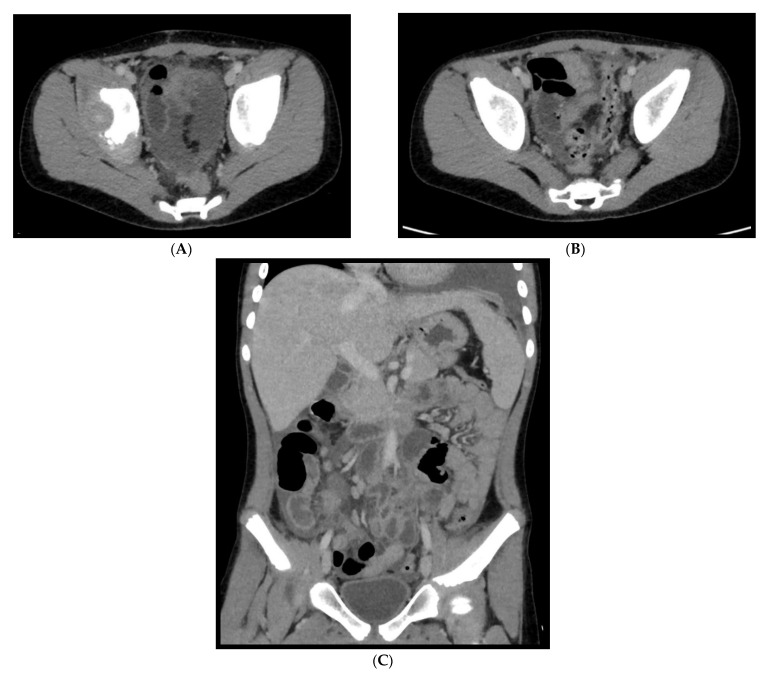
An 11-year-old patient with MIS-C. (**A**–**C**): Abdominal effusion with cecum and sigma hyper enhancing wall thickening.

**Table 1 microorganisms-09-02037-t001:** Adults “atypical” and “complicated” pulmonary COVID-19.

	Lymphadenopathy	Pleural Effusion	Pericardial Effusion	Bronchiectasis	Halo Sign/ Reverse Halo Sign	Cavitation	PNX*/ PNM*	
Authors								Total
Li et al. [8]	8.4%		4.8%					83
Grassi et al. [9]	54.8%	14.3%	16.7%					126
Xiao li et al. [10]	43.5%	14.3%						154
Zhou et al. [11]		3.2%						62
Darwish et al. [12]		13%						95
Ali Sabri et al. [13]			7.9%					63
Romain A. et al. [16]				41.3%				109
Devie A. et al. [17]				25.3%				158
Wu and Chen et al. [18]					13.8%/-			130
Bernheim et al. [19]					-/4%			121
Kuang et al. [20]					-/18.5%			405
Zoumut et al. [24]						11%		110
Vega et al. [25] (case rep.)							3 case of PNX and PNM	-
Loffi et al. [26]							-/6%	102

*PNX: pneumothorax; *PNM: pneumomediastinum.

**Table 2 microorganisms-09-02037-t002:** Children’s “Atypical” and “Complicated” Thoracic COVID-19 Manifestations.

	Lymphadenophaty	Pleural Effusion	Atelectasis	Nodules/ Linear Opacities	PNX/ PNM	
Authors						Total
Dominguez et al. [27]	17%	7%	2%	25%/33%	2%/-	91
Zhang et al. [29]	18%	0%				41
Charcape et al. [30]	0%	13%	2%	9.7%/15.6%		140

**Table 3 microorganisms-09-02037-t003:** Adults “Abdominal” COVID-19.

	Acute Aortic Occlusion (AAO)	Kidney/Splenic/Intestinal Infarction	Pancreatitis	Bowel Thickening (Colitis/Enteritis)	Cystitis	Cholecystitis and Biliary System Manifestation	
Authors							Total
Goldberg Stein et al. [46]		5%/5%/-		15%	5%	25%	80
Stacey Funt et al. [47]			1.5%	6.8%	4.1%		338
Bhayana et al. [48]		2.4%		29%			42
Baeza et al. [53] (case rep)	3 cases						-
Sree Tirumani et al [54]		1.3%/-/1.3%		1.3%			72

**Table 4 microorganisms-09-02037-t004:** Pediatric “Abdominal” COVID-19 Manifestation.

	GI Involvement	Hepato-Biliary Involvement	Pancreatic Involvement	
Authors				Total
Miller et al. [63]	20%			15
Tullie et al. [65] (case rep)	5 cases			-
Cantor et al. [73]		11%		44
Samie et al. [75] (case rep)			3 cases	-

## Data Availability

The authors confirm that the data supporting the findings of this review are available within the article and its references list.

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
