# Peer review of "The Multifaceted COVID-19: CT Aspects of Its Atypical Pulmonary and Abdominal Manifestations and Complications in Adults and Children. A Pictorial Review"

_microorganisms, 2021, doi:10.3390/microorganisms9102037_

Round 1

Reviewer 1 Report

This is a very interesting review regarding various and atypical radiological aspects of Covid-19, including even a description of the multisystemic radiological findings of the disease.

The work is very interesting, original and well-written.

I suggest to authors to provide a table to resume all the studies that analysed the radiological aspects described by authors.

If possible, it would be interesting adding a brief description of the role of ultrasound in Covid-19 (in particular pulmonary Covid-19).

Reviewer 2 Report

Authors delineated the CT finding of COVID-19, and provided typical and good qualities of CT pictures to support the document. However, the authors classified the CT findings by specific radiological terms, such as lymphadenopathy, pleural effusion, pericardiac effusion, etc. The findings were also presented in the complicated course of community acquired pneumonia, not limited to COVID-19 infection (https://doi.org/10.1016/S0012-3692(15)34476-7). There was little helpful information about clinical care of COVID-19. The authors presented abdominal complications and MIS-C. It would be better to explain the characteristics of CT findings in MIS-C, and further the association with lone COVID syndrome.

Round 2

Reviewer 2 Report

After revision, I think the manuscript was improving and readable. In current status, the manuscript would be accepted after minor English minor check.By the way, the Table 3 could be deleted. 
